# Differentially Private Covariance Estimation

**Kareem Amin**
kamin@google.com
Google Research NY

**Travis Dick**
tdick@cs.cmu.edu
Carnegie Mellon University

**Alex Kulesza**
kulesza@google.com
Google Research NY

**Andrés Muñoz Medina**
ammedina@google.com
Google Research NY

**Sergei Vassilvitskii**
sergeiv@google.com
Google Research NY

## Abstract

The task of privately estimating a covariance matrix is a popular one due to its applications to regression and PCA. While there are known methods for releasing private covariance matrices, these algorithms either achive only $(\epsilon, \delta)$-differential privacy or require very complicated sampling schemes, ultimately performing poorly in real data. In this work we propose a new $\epsilon$-differentially private algorithm for computing the covariance matrix of a dataset that addresses both of these limitations. We show that it has lower error than existing state-of-the-art approaches, both analytically and empirically. In addition, the algorithm is significantly less complicated than other methods and can be efficiently implemented with rejection sampling.

## 1 Introduction

Differential privacy has emerged as a standard framework for thinking about user privacy in the context of large scale data analysis [Dwork et al., 2014a]. While differential privacy does not protect against all attack vectors, it does provide formal guarantees about possible information leakage. A key feature of differential privacy is its robustness to post-processing: once a mechanism is certified as differentially private, arbitrary post-processing can be performed on its outputs without additional privacy impact.

The past decade has seen the emergence of a wide range of techniques for modifying classical learning algorithms to be differentially private [McSherry and Mironov, 2009, Chaudhuri et al., 2011, Jain et al., 2012, Abadi et al., 2016]. These algorithms typically train directly on the raw data, but inject carefully designed noise in order to produce differentially private outputs. A more general (and challenging) alternative approach is to first preprocess the dataset using a differentially private mechanism and then freely choose among standard off-the-shelf algorithms for learning. This not only provides more flexibility in the design of the learning system, but also removes the need for access to sensitive raw data (except for the initial preprocessing step). This approach thus falls under the umbrella of *data release*: since the preprocessed dataset is differentially private, it can, in principle, be released without leaking any individual's data.

In this work we consider the problem of computing, in a differentially private manner, a specific preprocessed representation of a dataset: its covariance matrix. Formally, given a data matrix $\mathbf{X} \in \mathbb{R}^{d \times n}$, where each column corresponds to a data point, we aim to compute a private estimate of $\mathbf{C} = \mathbf{X}\mathbf{X}^\top$ that can be used in place of the raw data, for example, as the basis for standard linear regression algorithms. Our methods provide privacy guarantees for the columns of $\mathbf{X}$.

There are many existing techniques that can be applied to this problem. We distinguish $\epsilon$-differentially private algorithms, which promise what is sometimes referred to as *pure differential*

*privacy*, from $(\epsilon, \delta)$-differentially private algorithms, which may fail to preserve privacy with some probability $\delta$. While algorithms in the pure differential privacy setting give stronger privacy guarantees, they tend to be significantly more difficult to implement, and often underperform empirically when compared to the straightforward algorithms in the $(\epsilon, \delta)$ setting.

In this work, we give a new practical $\epsilon$-differentially private algorithm for covariance matrix estimation. At a high level, the algorithm is natural. It approximates the eigendecomposition of the covariance matrix $\mathbf{C}$ by estimating the collections of eigenvalues and eigenvectors separately. Since the eigenvalues are insensitive to changes in a single column of $\mathbf{X}$, we can accurately estimate them using the Laplace mechanism. To estimate the eigenvectors, the algorithm uses the exponential mechanism to sample a direction $\theta$ from the unit sphere that approximately maximizes $\theta^\top \mathbf{C}\theta$, subject to the constraint of being orthogonal to the approximate eigenvectors sampled so far. The overall privacy guarantee for the combined method then follows from basic composition.

Our empirical results demonstrate lower reconstruction error for our algorithm when compared to other methods on both simulated and real-world datasets. This is especially striking in the high-privacy/low-$\epsilon$ regime, where we outperform all existing methods. We note that there is a different regime where our bounds no longer compete with those of the Gaussian mechanism, namely when $\epsilon$, $\delta$, and the number of data points are all sufficiently large (i.e., when privacy is "easy"). This suggests a two-pronged approach for the practitioner: utilize simple perturbation techniques when the data is insensitive to any one user and privacy parameters are lax, and more careful reconstruction when the privacy parameters are tight or the data is scarce, as is often the case in the social sciences and medical research.

Our main results can be summarized as follows:

- We prove our algorithm improves the privacy/utility trade-off by achieving lower error at a given privacy parameter compared with previous pure differentially private approaches (Theorem 2).
- We derive a non-uniform allocation of the privacy budget for estimating the eigenvectors of the covariance matrix giving the strongest utility guarantee from our analysis (Corollary 1).
- We show that our algorithm is practical: a simple rejection sampling scheme can be used for the core of the implementation (Algorithm 2).
- Finally, we perform an empirical evaluation of our algorithm, comparing it to existing methods on both synthetic and real-world datasets (Section 4). To the best of our knowledge, this is the first comparative empirical evaluation of different private covariance estimation methods, and we show that our algorithm outperforms all of the baselines, especially in the high privacy regime.

## 1.1 Database Sanitization for Ridge Regression

Our motivation for private covariance estimation is training regression models. In practice, regression models are trained using different subsets of features, multiple regularization parameters, and even varying target variables. If we were to directly apply differentially private learning algorithms for each of these learning tasks, our privacy costs would accumulate with every model we trained. Our goal is to instead pay the privacy cost only once, computing a single data structure that can be used multiple times to tune regression models. In this section, we show that a private estimate of the covariance matrix $\mathbf{C} = \mathbf{X}\mathbf{X}^\top$ summarizes the data sufficiently well for all of these ridge regression learning tasks with only a one-time privacy cost. Therefore, we can view differentially private covariance estimation as a database sanitization scheme for ridge regression.

Formally, given a data matrix $\mathbf{X} \in \mathbb{R}^{d \times n}$ with columns $x_1, \ldots, x_n$, we denote the $i^{\text{th}}$ entry of $x_j$ by $x_j(i)$. Consider using ridge regression to learn a linear model for estimating some target feature $x(t)$ as a function of $x(-t)$, where $x(-t)$ denotes the vector obtained by removing the $t^{\text{th}}$ feature of $x \in \mathbb{R}^d$. That is, we want solve the following regularized optimization problem:

$$w_\alpha = \operatorname*{argmin}_{w \in \mathbb{R}^{d-1}} \frac{1}{n} \sum_{j=1}^{n} \frac{1}{2}\left(w^\top x_j(-t) - x_j(t)\right)^2 + \alpha\|w\|_2^2.$$

We can write the solution to the ridge regression problem in closed form as follows. Let $\mathbf{A} \in \mathbb{R}^{(d-1) \times n}$ be the matrix consisting all but the $t^{\text{th}}$ row of $\mathbf{X}$ and $y = \left(x_1(t), \ldots, x_n(t)\right) \in \mathbb{R}^n$

be the $t^{\text{th}}$ row of $\mathbf{X}$ (as a column vector). Then the solution to the ridge regression problem with regularization parameter $\alpha$ is given by $w_\alpha = (\mathbf{A}\mathbf{A}^\top + 2\alpha n\mathbf{I})^{-1}\mathbf{A}y$.

Given access to just the covariance matrix $\mathbf{C} = \mathbf{X}\mathbf{X}^\top$, we can compute the above closed form ridge regression model. Suppose first that the target feature is $t = d$. Then, writing $\mathbf{X}$ in block-form, we have

$$\mathbf{C} = \mathbf{X}\mathbf{X}^\top = \begin{bmatrix} \mathbf{A} \\ y^\top \end{bmatrix} \begin{bmatrix} \mathbf{A}^\top & y \end{bmatrix} = \begin{bmatrix} \mathbf{A}\mathbf{A}^\top & \mathbf{A}y \\ y^\top\mathbf{A}^\top & y^\top y \end{bmatrix}.$$

Now it is not hard to see we can recover $w_\alpha$ by using the block entries of the full covariance matrix. The following lemma quantifies how much the error of estimating $\mathbf{C}$ privately affects the regression solution $w_\alpha$. The proof can be found in Appendix A.

**Lemma 1.** *Let $\mathbf{X} \in \mathbb{R}^{d \times n}$ be a data matrix, $\mathbf{C} = \mathbf{X}\mathbf{X}^\top \in \mathbb{R}^{d \times d}$, and $\hat{\mathbf{C}} \in \mathbb{R}^{d \times d}$ be a symmetric approximation to $\mathbf{C}$. Fix any target feature $t$ and regularization parameter $\alpha$. Let $w_\alpha$ and $\hat{w}_\alpha$ be the ridge regression models learned for predicting feature $t$ from $\mathbf{C}$ and $\hat{\mathbf{C}}$, respectively. Then*

$$\|w_\alpha - \hat{w}_\alpha\|_2 \leq \frac{\|\mathbf{C} - \hat{\mathbf{C}}\|_{2,\infty} + \|\mathbf{C} - \hat{\mathbf{C}}\|_2 \cdot \|\hat{w}_\alpha\|_2}{\lambda_{\min}(\mathbf{C}) + 2\alpha n},$$

*where $\|\mathbf{M}\|_{2,\infty}$ denotes the $L_{2,\infty}$-norm of $\mathbf{M}$ (the maximum 2-norm of its columns).*

Both $\|\mathbf{C} - \hat{\mathbf{C}}\|_{2,\infty}$ and $\|\mathbf{C} - \hat{\mathbf{C}}\|_2$ are upper bounded by the Frobenius error $\|\mathbf{C} - \hat{\mathbf{C}}\|_F$. Therefore, in our analysis of our differentially private covariance estimation mechanism, we will focus on bounding the Frobenius error. The bound in Lemma 1 also holds with $\|\hat{w}_\alpha\|_2$ replaced by $\|w_\alpha\|_2$ in the right hand side, however we prefer the stated version since it can be computed by the practitioner.

## 1.2 Related Work

A variety of techniques exist for computing differentially private estimates of covariance matrices, including both general mechanisms that can be applied in this setting as well as specialized methods that take advantage of problem-specific structure.

A naïve approach using a standard differential privacy mechanism would be to simply add an appropriate amount of Laplace noise independently to every element in the true covariance matrix $\mathbf{C}$. However, the amount of noise required makes such a mechanism impractical, as the sensitivity, and hence the amount of noise added, grows linearly in the dimension. A better approach is to add Gaussian noise [Dwork et al., 2014b]; however, this results in $(\epsilon, \delta)$-differential privacy, where, with some probability $\delta$, the outcome is not private. Similarly, Upadhyay [2018] proposes a private way of generating low dimensional representations of $\mathbf{X}$. This is a slightly different task than covariance estimation. Moreover, their algorithm is only $(\epsilon, \delta)$-differentially private for $\delta > n^{-\log n}$ which makes the privacy regime incomparable to the one proposed in this paper. Another approach, proposed in Chaudhuri et al. [2012], is to compute a private version of PCA. This approach has two limitations. First, it only works for computing the top eigenvectors, and can fail to give non-trivial results for computing the full covariance matrix. Second, the sampling itself is quite involved and requires the use of a Gibbs sampler. Since it is generally impossible to know when the sampler converges, adding noise in this manner can violate privacy guarantees.

The algorithm we propose bears the most resemblance to the differentially private low-rank matrix approximation proposed by Kapralov and Talwar [2013], which approximates the SVD. Their algorithm computes a differentially private rank-1 approximation of a matrix $C$, subtracts this matrix from $C$ and then iterates the process on the residual. Similarly, our approach iteratively generates estimates of the eigenvectors of the matrix, but repeatedly *projects* the matrix onto the subspace orthogonal to the previously estimated eigenvectors. We demonstrate the benefit of this projective update both in our analytical bounds and empirical results. This ultimately allows us to rely on a simple rejection sampling technique proposed by Kent et al. [2018] to select our eigenvectors.

Other perturbation approaches include recent work on estimating sparse covariance matrices by Wang and Xu [2019]. Their setup differs from ours in that they assume all columns in the covariance matrix have $s$-sparsity. There was also an attempt by Jiang et al. [2016] to use Wishart-distributed noise to privately estimate a covariance matrix. However, Imtiaz and Sarwate [2016] proposed the same algorithm and later discovered that the algorithm was in fact not differentially private.

Wang [2018] also study the effectiveness of differentially private covariance estimation for private linear regression (and compare against several other private regression approaches). However, they only consider the Laplace and Gaussian mechanisms for private covariance estimation and do not study the quality of the estimated covariance matrices, only their performance for regression tasks.

## 2 Preliminaries

Let $\mathbf{X} \in \mathbb{R}^{d \times n}$ be a data matrix where each column corresponds to a $d$-dimensional data point. Throughout the paper, we assume that the columns of the data matrix have $\ell_2$-norm at most one[1]. Our goal is to privately release an estimate of the unnormalized and uncentered covariance matrix $\mathbf{C} = \mathbf{X}\mathbf{X}^\top \in \mathbb{R}^{d \times d}$.

We say that two data matrices $\mathbf{X}$ and $\tilde{\mathbf{X}}$ are neighbors if they differ on at most one column, denoted by $\mathbf{X} \sim \tilde{\mathbf{X}}$. We want algorithms that are $\epsilon$-differentially private with respect to neighboring data matrices. Formally, an algorithm $\mathcal{A}$ is $\epsilon$-differentially private if for every pair of neighboring data matrices $\mathbf{X}$ and $\tilde{\mathbf{X}}$ and every set $\mathcal{O}$ of possible outcomes, we have:

$$\Pr(\mathcal{A}(\mathbf{X}) \in \mathcal{O}) \leq e^\epsilon \Pr(\mathcal{A}(\tilde{\mathbf{X}}) \in \mathcal{O}) . \tag{1}$$

A useful consequence of this definition is *composability*.

**Lemma 2.** *Suppose an algorithm $\mathcal{A}_1 : \mathbb{R}^{d \times n} \to \mathcal{Y}_1$ is $\epsilon_1$-differentially private and a second algorithm $\mathcal{A}_2 : \mathbb{R}^{d \times n} \times \mathcal{Y}_1 \to \mathcal{Y}_2$ is $\epsilon_2$-differentially private. Then the composition $\mathcal{A}(X) = \mathcal{A}_2(X, \mathcal{A}_1(X))$ is $(\epsilon_1 + \epsilon_2)$-differentially private.*

Our main algorithm uses this property and multiple applications of the following mechanisms.

**Laplace Mechanism.** Let $\mathrm{Lap}(\alpha)$ denote the Laplace distribution with parameter $\alpha$. Given a query $f : \mathbb{R}^{d \times n} \to \mathbb{R}^k$ mapping data matrices to vectors, the $\ell_1$-sensitivity of the query is given by $\Delta_f = \max_{\mathbf{X} \sim \tilde{\mathbf{X}}} \|f(\mathbf{X}) - f(\tilde{\mathbf{X}})\|_1$. For a given privacy parameter $\epsilon$, the *Laplace mechanism* approximately answers queries by outputting $f(\mathbf{X}) + (Y_1, \ldots, Y_k)$, where each $Y_i$ is independently sampled from the $\mathrm{Lap}(\Delta_f/\epsilon)$ distribution. The privacy and utility guarantees of the Laplace mechanism are summarized in the following lemma.

**Lemma 3.** *The Laplace mechanism preserves $\epsilon$-differential privacy and, for any $\beta > 0$, we have $\Pr(\max_i |Y_i| \geq \frac{\Delta_f}{\epsilon} \log \frac{k}{\beta}) \leq \beta$.*

**Exponential Mechanism.** The exponential mechanism can be used to privately select an approximately optimal outcome from an arbitrary domain. Formally, let $(\mathcal{Y}, \mu)$ be a measure space and $g : (\mathbf{X}, y) \mapsto g(\mathbf{X}, y)$ be the utility of outcome $y$ for data matrix $\mathbf{X}$. The sensitivity of $g$ is given by $\Delta_g = \max_{\mathbf{X} \sim \tilde{\mathbf{X}}, y} |g(\mathbf{X}, y) - g(\tilde{\mathbf{X}}, y)|$. For $\epsilon > 0$, the exponential mechanism samples $y$ from density proportional to $f_{\exp}(y) = \exp(\frac{\epsilon}{2\Delta_g} g(\mathbf{X}, y))$, defined with respect to the base measure $\mu$.

**Lemma 4** (McSherry and Talwar [2007])**.** *The exponential mechanism preserves $\epsilon$-differential privacy. Let $OPT = \max_y g(\mathbf{X}, y)$ and $G_\tau = \{y \in \mathcal{Y} : g(\mathbf{X}, y) \geq OPT - \tau\}$. If $\hat{y}$ is the output of the exponential mechanism, we have $\Pr(\hat{y} \notin G_{2\tau}) \leq \exp(-\epsilon\tau/(2\Delta_g)) \cdot \mu(G_\tau)$.*

In our algorithm, we will apply the exponential mechanism in order to choose unit-length approximate eigenvectors. Therefore, the space of outcomes $\mathcal{Y}$ will be the unit sphere $\mathcal{S}^{d-1} = \{\theta \in \mathbb{R}^d : \|\theta\|_2 = 1\}$. For convenience, we will use the uniform distribution on the sphere, denoted by $\mu_\circ$, as our base measure (this is proportional to the surface area). For example, the density $p(\theta) = 1$ is the uniform distribution on $\mathcal{S}^{d-1}$ and $\mu_\circ(\mathcal{S}^{d-1}) = 1$.

## 3 Iterative Eigenvalue Sampling for Covariance Estimation

In this section we describe our $\epsilon$-differentially private covariance estimation mechanism. In fact, our method produces a differentially private approximation to the *eigendecomposition* of $\mathbf{C} = \mathbf{X}\mathbf{X}^\top$.

We first estimate the vector of eigenvalues, a query that has $\ell_1$-sensitivity at most 2. Next, we show how to use the exponential mechanism to approximate the top eigenvector of the covariance matrix $\mathbf{C}$. Inductively, after estimating the top $k$ eigenvectors $\hat{\theta}_1, \ldots, \hat{\theta}_k$ of $\mathbf{C}$, we project the data onto the $(d-k)$-dimensional orthogonal subspace and apply the exponential mechanism to approximate the top eigenvector of the remaining projected covariance matrix. Once all eigenvalues and eigenvectors have been estimated, the algorithm returns the reconstructed covariance matrix. Pseudocode for our method is given in Algorithm 1. In Section 3.2 we discuss a rejection-sampling algorithm of Kent et al. [2018] that can be used for sampling the distribution defined in step (a) of Algorithm 1. It is worth mentioning that if we only sample $k$ eigenvectors, Algorithm 1 would return a rank k-approximation of matrix $\mathbf{C}$.

---

**Algorithm 1** Iterative Eigenvector Sampling

---

**Input:** $\mathbf{C} = \mathbf{X}\mathbf{X}^\top \in \mathbb{R}^{d \times d}$, privacy parameters $\epsilon_0, \ldots, \epsilon_d$.

1. Initialize $\mathbf{C}_1 = \mathbf{C}$, $\mathbf{P}_1 = \mathbf{I} \in \mathbb{R}^{d \times d}$, $\hat{\lambda}_i = \lambda_i(\mathbf{C}) + \mathrm{Lap}(2/\epsilon_0)$ for $i = 1, \ldots, d$.

2. For $i = 1, \ldots, d$:

    (a) Sample $\hat{u}_i \in \mathcal{S}^{d-i}$ proportional to $f_{\mathbf{C}_i}(u) = \exp(\frac{\epsilon_i}{4} u^\top \mathbf{C}_i u)$ and let $\hat{\theta}_i = \mathbf{P}_i^\top \hat{u}_i$.

    (b) Find an orthonormal basis $\mathbf{P}_{i+1} \in \mathbb{R}^{(d-i) \times d}$ orthogonal to $\hat{\theta}_1, \ldots, \hat{\theta}_i$.

    (c) Let $\mathbf{C}_{i+1} = \mathbf{P}_{i+1} \mathbf{C} \mathbf{P}_{i+1}^\top \in \mathbb{R}^{(d-i) \times (d-i)}$.

3. Output $\hat{\mathbf{C}} = \sum_{i=1}^d \hat{\lambda}_i \hat{\theta}_i \hat{\theta}_i^\top$.

---

Our approach is similar to the algorithm of Kapralov and Talwar [2013] with one significant difference: in their algorithm, rather than projecting onto the orthogonal subspace of the first $k$ estimated eigenvectors, they subtract the rank-one matrix given by $\hat{\lambda}_i \hat{\theta}_i \hat{\theta}_i^\top$ from $\mathbf{C}$, where $\hat{\lambda}_i$ is the estimate of the $i^{\text{th}}$ eigenvalue. There are several advantages to using projections. First, the projection step exactly eliminates the variance along the direction $\hat{\theta}_i$, while the rank-one subtraction will fail to do so if the estimated eigenvalues are incorrect (effectively causing us to pay for the eigenvalue approximation twice: once in the reconstruction of the covariance matrix and once because it prevents us from removing the variance along the direction $\hat{\theta}_i$ before estimating the remaining eigenvectors). Second, the analysis of the algorithm is substantially simplified because we are guaranteed that the estimated eigenvectors $\hat{\theta}_1, \ldots, \hat{\theta}_d$ are orthogonal, and we do not require bounds for rank-one updates on the spectrum of a matrix.

We now show that Algorithm 1 is differentially private. The algorithm applies the Laplace mechanism once and the exponential mechanism $d$ times, so the result follows from bounding the sensitivity of the relevant queries and applying basic composition.

**Theorem 1.** *Algorithm 1 preserves $\left(\sum_{i=0}^d \epsilon_i\right)$-differential privacy.*

We now focus on the main contribution of this paper: a utility guarantee for Algorithm 1 in terms of the Frobenius distance between $\hat{\mathbf{C}}$ and the true covariance matrix $\mathbf{C}$ as a function of the privacy parameters used for each step. An important consequence of this analysis is that we can optimize the allocation of our total privacy budget $\epsilon$ among the $d+1$ queries in order to get the best bound.

First we provide a utility guarantee for the exponential mechanism applied to approximating the top eigenvector of a matrix $\mathbf{C}$. This result is similar to the rank-one approximation guarantee given by Kapralov and Talwar [2013], but we include a proof in the appendix for completeness.

**Lemma 5.** *Let $\mathbf{X} \in \mathbb{R}^{d \times n}$ be a data matrix and $\mathbf{C} = \mathbf{X}\mathbf{X}^\top$. For any $\beta > 0$, with probability at least $1 - \beta$ over $\hat{u}$ sampled from the density proportional to $f_{\mathbf{C}}(u) = \exp(\frac{\epsilon}{4} u^\top \mathbf{C} u)$ on $\mathcal{S}^{d-1}$, we have*

$$\hat{u}^\top \mathbf{C} \hat{u} \geq \lambda_1(\mathbf{C}) - O\left(\frac{1}{\epsilon}\left(d \log \lambda_1(\mathbf{C}) + \log \frac{1}{\beta}\right)\right)$$

The following result characterizes the dependence of the Frobenius error on the errors in the estimated eigenvalues and eigenvectors. In particular, given that the eigenvalue estimates all have bounded error, the dependence on the $i^{\text{th}}$ eigenvector estimate $\hat{\theta}_i$ is only through the quantity

$\lambda_i(\mathbf{C}) - \hat{\theta}_i^\top \mathbf{C} \hat{\theta}_i$, which measures how much less variance of $\mathbf{C}$ is captured by $\hat{\theta}_i$ as compared to the true $i^{\text{th}}$ eigenvector. Moreover, the contribution of $\hat{\theta}_i$ is roughly weighted by $\hat{\lambda}_i$. This observation allows us to tune the privacy budgeting across the $d$ eigenvector queries, allocating more budget (at runtime) to the eigenvectors with large estimated eigenvalues. Empirically, we find that this budget allocation step improves performance in some settings.

**Lemma 6.** *Let $\mathbf{C} \in \mathbb{R}^{d \times d}$ be any positive semidefinite matrix. Let $\hat{\theta}_1, \ldots, \hat{\theta}_d$ be any orthonormal vectors and $\hat{\lambda}_1, \ldots, \hat{\lambda}_d$ be estimates of the eigenvalues of $\mathbf{C}$ satisfying $|\hat{\lambda}_i - \lambda_i(\mathbf{C})| \leq \tau$ for all $i \in [d]$. Then*

$$\|C - \hat{\Theta}\hat{\Lambda}\hat{\Theta}^\top\|_F \leq \sqrt{2 \sum_{i=1}^d \lambda_i(\mathbf{C}) \cdot (\lambda_i(\mathbf{C}) - \hat{\theta}_i^\top \mathbf{C} \hat{\theta}_i)} + \tau\sqrt{d}$$

*where $\hat{\Theta}$ is the matrix with columns $\hat{\theta}_i$ and $\hat{\Lambda}$ is the diagonal matrix with entries $\hat{\lambda}_i$.*

*Proof.* Let $\Lambda \in \mathbb{R}^{d \times d}$ be the diagonal matrix of true eigenvalues of $\mathbf{C}$. We have $\|\mathbf{C} - \hat{\Theta}\hat{\Lambda}\hat{\Theta}^\top\|_F \leq \|\mathbf{C} - \hat{\Theta}\Lambda\hat{\Theta}^\top\|_F + \|\hat{\Theta}(\Lambda - \hat{\Lambda})\hat{\Theta}^\top\|_F$. The second term is bounded by $\tau\sqrt{d}$, so it remains to bound the first term. We have that

$$\|\mathbf{C} - \hat{\Theta}\Lambda\hat{\Theta}^\top\|_F^2 = \|\mathbf{C}\|_F^2 + \|\hat{\Theta}\Lambda\hat{\Theta}^\top\|_F^2 - 2\operatorname{tr}(\mathbf{C}\hat{\Theta}\Lambda\hat{\Theta}^\top) = 2\sum_i \lambda_i(\mathbf{C})^2 - 2\sum_i \lambda_i(\mathbf{C})\hat{\theta}_i^\top \mathbf{C}\hat{\theta}_i$$

$$= 2\sum_i \lambda_i(\mathbf{C})(\lambda_i(\mathbf{C}) - \hat{\theta}_i^\top \mathbf{C}\hat{\theta}_i),$$

where the second equation follows from the fact that the first two terms are both equal to $\sum_i \lambda_i(\mathbf{C})^2$ and the cyclic property of the trace. The final bound follows by taking the square root. $\square$

We are now ready to prove our main utility guarantee for Algorithm 1. The remaining analysis focuses on the effect of working with the projected covariance matrices $\mathbf{C}_i$. One interesting observation is that our algorithm does not have error accumulating across its iterations due to the projection step. Following Lemma 6, we only need to show that $\hat{\theta}_i$ captures nearly as much of the variance of $\mathbf{C}$ as the $i^{\text{th}}$ eigenvector. Fortunately, if our estimates $\hat{\theta}_1, \ldots, \hat{\theta}_{i-1}$ have errors, then the orthogonal subspace only contains *more* variance, and thus the sampling step in round $i$ actually becomes easier. In this sense Algorithm 1 is "self-correcting".

**Theorem 2.** *Let $\hat{\mathbf{C}}$ be the output of Algorithm 1 run with inputs $\mathbf{C}$ and privacy parameters $\epsilon_0, \ldots, \epsilon_d$. For any $\beta > 0$, with probability at least $1 - \beta$ we have*

$$\|\mathbf{C} - \hat{\mathbf{C}}\|_F \leq \tilde{O}\left(\sqrt{\sum_{i=1}^d \frac{d\lambda_i(\mathbf{C})}{\epsilon_i}} + \frac{\sqrt{d}}{\epsilon_0}\right),$$

*where the $\tilde{O}$ notation suppresses logarithmic terms in $d$, $\lambda_1(\mathbf{C})$, and $\beta$.*

If Algorithm 1 is used to obtain a $k$-rank approximation, the above theorem can be modified to show that the distance from the best $k$-rank approximation would be in $O\left(\sqrt{\sum_{i=1}^k \frac{d\lambda_i(\mathbf{C})}{\epsilon_i}} + \frac{\sqrt{d}}{\epsilon_0}\right)$. Since Theorem 2 bounds the error in terms of the privacy parameters $\epsilon_0, \ldots, \epsilon_d$, we can tune our allocation of the total privacy budget of $\epsilon$ across the $d + 1$ private operations in order to obtain the tightest possible bound. In order to preserve privacy, we tune based on the estimated eigenvalues $\hat{\lambda}_1, \ldots, \hat{\lambda}_d$ obtained in step (1) of Algorithm 1 rather than using the true eigenvalues. The following result makes precise the natural intuition that more effort should be made to estimate those eigenvectors with larger (estimated) eigenvalues; its proof can be found in Appendix B.

**Corollary 1.** *Fix any privacy parameter $\epsilon$ and any failure probability $\beta > 0$, let $\epsilon_0 = \epsilon/2$, and let $\epsilon_i = \frac{\frac{\epsilon}{2}\sqrt{\hat{\lambda}_i + \tau}}{\sum_j \sqrt{\hat{\lambda}_j + \tau}}$ where $\tau = \frac{2}{\epsilon_0}\log(2d/\beta)$. Then Algorithm 1 run with $\epsilon_0, \ldots, \epsilon_d$ preserves $\epsilon$-differential privacy and, with probability at least $1 - \beta$, the output $\hat{\mathbf{C}}$ satisfies*

$$\|\mathbf{C} - \hat{\mathbf{C}}\|_F \leq \tilde{O}\left(\sqrt{\frac{d}{\epsilon}} \sum_{i=1}^d \sqrt{\hat{\lambda}_i + \frac{1}{\epsilon}} + \frac{\sqrt{d}}{\epsilon}\right).$$

## 3.1 Comparison of Bounds

In this section we compare the bound provided by Theorem 2 to previous state-of-the-art results.

**Comparison to Kapralov and Talwar [2013].** The bounds given by Kapralov and Talwar [2013], when applied to the case of recovering the full-rank covariance matrix, bound the spectral error $\|\mathbf{C} - \hat{\mathbf{C}}\|_2$ by $\zeta\lambda_1(\mathbf{C})$ (for some $\zeta > 0$) under the condition that $\lambda_1(\mathbf{C})$ is sufficiently large. In particular, Theorem 18 from their paper shows that there exists an $\epsilon$-differentially private algorithm with the above guarantee whenever $\lambda_1(\mathbf{C}) \geq C_1 d^4/(\epsilon\zeta^6)$ for some constant $C_1$. Since $\|\mathbf{C} - \hat{\mathbf{C}}\|_2 \leq \|\mathbf{C} - \hat{\mathbf{C}}\|_F$, we can directly compare both algorithms after slightly rewriting our bounds. The following result shows that we improve the necessary lower bound on $\lambda_1(\mathbf{C})$ by a factor of $d/\zeta^4$ (ignoring log terms).

**Corollary 2.** *For any $\zeta > 0$ and any positive semidefinite matrix $\hat{\mathbf{C}}$, with probability at least 0.99 (or any fixed success probability), running Algorithm 1 with $\epsilon_0 = \epsilon/2$ and $\epsilon_i = \epsilon/(2d)$ for $i = 1, \ldots, d$ preserves $\epsilon$-differential privacy and outputs $\hat{\mathbf{C}}$ such that $\|\mathbf{C} - \hat{\mathbf{C}}\|_F \leq O(\zeta\lambda_1(\mathbf{C}))$ if $\lambda_1(\mathbf{C}) \geq \frac{2d^3}{\epsilon\zeta^2}\log(\frac{d}{\epsilon\zeta})$.*

**Comparison to Gaussian Mechanism.** We can also directly compare to the error bounds for the Gaussian mechanism given by Dwork et al. [2014b]. Theorem 9 in their paper gives $\|\mathbf{C} - \hat{\mathbf{C}}\|_F \leq O(d^{3/2}\sqrt{\log(1/\delta)}/\epsilon)$, where $\epsilon$ and $\delta$ are the (approximate) differential privacy parameters. Using privacy parameters $\epsilon_0 = \epsilon/2$ and $\epsilon_i = \epsilon/(2d)$ for $i = 1, \ldots, d$, Theorem 2 implies that with high probability we have $\|\mathbf{C} - \hat{\mathbf{C}}\|_F \leq O\big(d^{3/2}\sqrt{\lambda_1(\mathbf{C})\log(\lambda_1(\mathbf{C}))}/\epsilon + \sqrt{d}/\epsilon\big)$. For all values of $\delta > 0$, our algorithm provides a stronger privacy guarantee than the Gaussian mechanism. On the other hand, whenever $\lambda_1(\mathbf{C})\log(\lambda_1(\mathbf{C})) \leq \log(1/\delta)/\epsilon$, our utility guarantee is tighter. Given that $\lambda_1(\mathbf{C}) = O(n)$, where $n$ is the number of data points, we see that our algorithm admits better utility guarantees in both the low data regime and the high privacy regime.

## 3.2 Sampling on the Sphere

To implement Algorithm 1, we need a subprocedure for drawing samples from the densities proportional to $\exp(\frac{\epsilon}{4}u^\top\mathbf{C}u)$ defined on the sphere $\mathcal{S}^{d-1}$, where $\mathbf{C}$ is a covariance matrix and $\epsilon$ is the desired privacy parameter. This density belongs to a family called Bingham distributions. Kapralov and Talwar [2013] also discuss this sampling problem and, while their algorithm could also be used in our setting, we instead rely on a simpler rejection-sampling scheme proposed by Kent et al. [2018]. This sampling technique is exact and we find empirically that it is very efficient. Pseudocode for their method is given in Algorithm 2 in the appendix.

Recall that rejection sampling allows us to generate samples from the distribution with density proportional to $f$, provided we can sample from the distribution with density proportional to a similar function $g$, called the envelope. Kent et al. [2018] propose to use the angular central Gaussian distribution as an envelope. This distribution has a matrix parameter $\mathbf{\Omega}$ and unnormalized density (defined on the sphere $\mathcal{S}^{d-1}$) given by $g(u) = (u^\top\mathbf{\Omega}u)^{-d/2}$. To sample from this distribution, we can simply sample $z$ from the mean-zero Gaussian distribution with covariance given by $\mathbf{\Omega}^{-1}$ and output $u = z/\|z\|_2$. Kent et al. [2018] provide a choice of parameter $\mathbf{\Omega}$ to minimize the number of rejected samples. They show that under some reasonable assumptions the expected number of rejections grows like $O(\sqrt{d})$ (see [Kent et al., 2018] for more details). In our experiments we observed the median number of samples was less than $d$, and the mean was around $2d$. We believe that our empirical rejection counts are larger than the asymptotic bounds of Kent et al. [2018] because the dimensionality of our datasets is not large enough.

# 4 Experiments

We now present the results of an extensive empirical evaluation of the performance of our algorithm. Given a data matrix $\mathbf{X}$, we study the performance of the algorithm on two tasks: (i) privately estimating the covariance matrix $\mathbf{C} = \mathbf{X}\mathbf{X}^\top$, and (ii) privately regressing to predict one of the columns of $\mathbf{X}$ from the others. Due to space constraints we present only the results of (i) and present the results of (ii) in Appendix C.

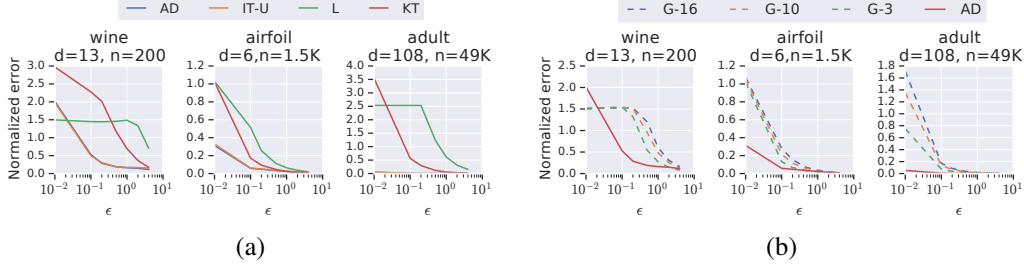

Figure 1: Results comparing our algorithm across the wine, airfoil and adult data sets. (a) Comparison to **KT** and **L**. Error is normalized Frobenius distance. (b) Comparison to the Gaussian mechanism. The legend G-$x$ corresponds to a value of $\delta = 10^{-x}$.

We compare the performance of our algorithm to a number of different baselines. We begin with two general purpose output perturbation methods: the Laplace mechanism and the Gaussian mechanism.

- The Laplace mechanism [Dwork et al., 2006] (**L**). The output is given by $\hat{\mathbf{C}} = \mathbf{C} + \mathbf{M}$ where $\mathbf{M}$ is a matrix with entries distributed $\mathrm{Lap}(\frac{2d}{\epsilon})$.

- The Gaussian mechanism [Dwork et al., 2014b] (**G**). Notably, the Gaussian mechanism achieves $(\epsilon, \delta)$-differential privacy, hence its privacy guarantees are weaker for the same value of $\epsilon$. Our goal is to measure if we can achieve similar utility under stricter privacy constraints. We experiment with different values of $\delta$.

- The algorithm proposed by Kapralov and Talwar [2013] (**KT**). This algorithm is $\epsilon$-differentially private. We use Algorithm 2 for the vector sampling subroutine.

- Algorithm 1 with adaptive privacy splitting (**AD**). We allocate the privacy budget in the manner suggested by Corollary 1.

- Algorithm 1 with uniform privacy splitting (**IT-U**). Same as above except the privacy budget used to sample eigenvectors is split uniformly.

One final modification we apply to all algorithms that release a covariance matrix is to round the eigenvalues of the private matrix to fall in the interval $[0, n]$, since this bound is data-independent and is easy to derive analytically.

We measure the performance of our algorithm on three different datasets: Wine, Adult, and Airfoil from the UCI repository[2], These datasets have dimensions ranging from 13 to 108, and number of points from 200 to 49,000. The approximation error of each algorithm is measured using the normalized Frobenius distance $\frac{\|\hat{\mathbf{C}} - \mathbf{C}\|_F}{n}$. To investigate the privacy/utility trade-off, we run each algorithm with privacy parameter $\epsilon \in \{0.01, 0.1, 0.2, 0.5, 1.0, 2.0, 4.0\}$. For the Gaussian mechanism, we also varied the parameter $\delta \in \{1e^{-16}, 1e^{-10}, 1e^{-3}\}$ We ran each experiment 50 times, showing the average error in Figure 1.

The first thing to notice is that our algorithm consistently outperforms all others except for the single case of the wine data set with $\epsilon = 0.01$. Recall that the Gaussian mechanism has an additional failure probability $\delta$, thus the privacy guarantees we obtain are strictly better for the same value of $\epsilon$. Therefore, it is particularly striking that we consistently beat the Gaussian mechanism even for the very relaxed value of $\delta = .001$.

Another important observation from this experiment is that the adaptive and non adaptive privacy budget splitting seems to not have a big effect on the performance of the algorithm. Finally, we see that the performance gap between **AD** and **KT** is largest on the dataset with the highest dimension. This phenomenon is in line with the analysis of Section 3.1. We explore this effect in more detail in Appendix C.

Finally, as we detail in Appendix C our approach outperforms the output perturbation method of Chaudhuri et al. [2011] on the regression task, even though the latter achieves $(\epsilon, \delta)$-differential privacy. As we mentioned previously, the private covariance matrix output by our algorithm can

be also be used to tune regularization parameters without affecting the privacy budget, thus giving additional freedom to practitioners in tuning their algorithms.

## 5   Conclusion

We presented a new algorithm for differentially private covariance estimation, studied it analytically, and demonstrated its performance on a number of synthetic and real world datasets. To the best of our knowledge this is the first $\epsilon$-differentially private algorithm to admit a utility guarantee that grows as $O(d^{3/2})$ with the dimension of the dataset. Previously, such bounds could only be achieved at the cost $(\epsilon, \delta)$-differential privacy. We also showed that the average Frobenius approximation error of our algorithm decreases as $O\left(\frac{1}{\sqrt{n}}\right)$, which is slower than the $O\left(\frac{1}{n}\right)$ rate of the Gaussian and Laplace mechanisms. This poses an open question of whether the suboptimal dependency on $n$ is necessary in order to achieve pure differential privacy or to achieve a dependency on the dimension of $O(d^{3/2})$.

Looking more broadly, practical machine learning and data analysis typically requires a significant amount of tuning: feature selection, hyperparameter selection, experimenting with regularization, and so on. If this tuning is performed using the underlying private dataset, then in principle all of these count against the privacy budget of the algorithm designer (who must also, of course, have access to that private dataset). By producing a differentially private *summary* of the dataset from which multiple models can be trained with no additional privacy cost, our approach allows a practitioner to operate freely, without worrying about privacy budgets or the secure handling of private data. We believe that finding techniques for computing private representations in other settings is an exciting direction for future research.

## Footnotes

[1]If the columns of $\mathbf{X}$ have $\ell_2$-norm bounded by a known value $B$, we can rescale the columns by $1/B$ to obtain a matrix $\mathbf{X}'$ with column norm at most 1. Since $\mathbf{X}\mathbf{X}^\top = B^2 \mathbf{X}'\mathbf{X}'^\top$, estimating the covariance matrix of $\mathbf{X}'$ gives an estimate of the covariance matrix of $\mathbf{X}$ with Frobenius error inflated by a factor $B^2$.

[2]https://archive.ics.uci.edu/ml/datasets/

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
