[Supplementary Material · 7986_supplement.pdf]

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

## A  Proofs for Regression Bounds

To simplify notation, we will let $\mathbf{A}(\mathbf{C}, t) \in \mathbb{R}^{(d-1)\times n}$ and $b(\mathbf{C}, t) \in \mathbb{R}^n$ denote the blocks extracted from $\mathbf{C}$ so that the ridge regression model for predicting feature $t$ with regularization parameter $\alpha$ is given by $w_\alpha = (\mathbf{A}(\mathbf{C}, t) + 2\alpha n \mathbf{I})^{-1} b(\mathbf{C}, t)$.

**Lemma 1.** *Let $\mathbf{X} \in \mathbb{R}^{d\times n}$ be a data matrix, $\mathbf{C} = \mathbf{X}\mathbf{X}^\top \in \mathbb{R}^{d\times d}$, and $\hat{\mathbf{C}} \in \mathbb{R}^{d\times d}$ be a symmetric approximation to $\mathbf{C}$. Fix any target feature $t$ and regularization parameter $\alpha$. Let $w_\alpha$ and $\hat{w}_\alpha$ be the ridge regression models learned for predicting feature $t$ from $\mathbf{C}$ and $\hat{\mathbf{C}}$, respectively. Then*

$$\|w_\alpha - \hat{w}_\alpha\|_2 \leq \frac{\|\mathbf{C} - \hat{\mathbf{C}}\|_{2,\infty} + \|\mathbf{C} - \hat{\mathbf{C}}\|_2 \cdot \|\hat{w}_\alpha\|_2}{\lambda_{\min}(\mathbf{C}) + 2\alpha n},$$

*where $\|\mathbf{M}\|_{2,\infty}$ denotes the $L_{2,\infty}$-norm of $\mathbf{M}$ (the maximum 2-norm of its columns).*

*Proof.* First, we introduce some shorthand notation: let $\mathbf{A} = \mathbf{A}(\mathbf{C}, t)$, $b = b(\mathbf{C}, t)$, $\hat{\mathbf{A}} = \mathbf{A}(\hat{\mathbf{C}}, t)$, and $\hat{b} = b(\hat{\mathbf{C}}, t)$ be the blocks extracted from $\mathbf{C}$ and $\hat{\mathbf{C}}$ so that $(\mathbf{A} + 2\alpha n \mathbf{I})w_\alpha = b$ and $(\hat{\mathbf{A}} + 2\alpha n \mathbf{I})\hat{w}_\lambda = \hat{b}$. Subtracting the equalities defining $\hat{w}_\alpha$ and $w_\alpha$, we have

$$(\hat{\mathbf{A}} + 2\alpha n \mathbf{I})\hat{w}_\alpha - (\mathbf{A} + 2\alpha n \mathbf{I})w_\alpha = \hat{b} - b.$$

Expanding the left hand side and adding and subtracting $\mathbf{A}\hat{w}_\alpha$, we have the following equivalences

$$(\hat{\mathbf{A}} + 2\alpha n \mathbf{I})\hat{w}_\alpha - (\mathbf{A} + 2\alpha n \mathbf{I})w_\alpha = \hat{b} - b$$
$$\iff \hat{\mathbf{A}}\hat{w}_\alpha - \mathbf{A}\hat{w}_\alpha + \mathbf{A}\hat{w}_\alpha - \mathbf{A}w_\alpha + 2\alpha n \mathbf{I}(\hat{w}_\alpha - w_\alpha) = \hat{b} - b$$
$$\iff \hat{w}_\alpha - w_\alpha = (\mathbf{A} + 2\alpha n \mathbf{I})^{-1}\big(\hat{b} - b - (\hat{\mathbf{A}} - \mathbf{A})\hat{w}_\alpha\big).$$

Therefore

$$\|\hat{w}_\alpha - w_\alpha\|_2 \leq \|(\mathbf{A} + 2\alpha n \mathbf{I})^{-1}\|_2 (\|\hat{b} - b\|_2 + \|\hat{\mathbf{A}} - \mathbf{A}\|_2 \cdot \|\hat{w}_\alpha\|_2)$$
$$= \frac{\|\hat{b} - b\|_2 + \|\hat{\mathbf{A}} - \mathbf{A}\|_2 \cdot \|\hat{w}_\alpha\|_2}{\lambda_{\min}(\mathbf{A}) + 2\alpha n}.$$

It remains to show that $\|\hat{b} - b\|_2 \leq \|\hat{\mathbf{C}}\|_{2,\infty}$, $\|\hat{\mathbf{A}} - \mathbf{A}\|_2 \leq \|\hat{\mathbf{C}} - \mathbf{C}\|_2$, and $\lambda_{\min}(\mathbf{A}) \geq \lambda_{\min}(\mathbf{C})$.

First, since $b$ and $\hat{b}$ correspond to the $t^{\text{th}}$ columns of $\mathbf{C}$ and $\hat{\mathbf{C}}$ after removing the $t^{\text{th}}$ entries, we have that $\|\hat{b} - b\|_2 \leq \|\hat{\mathbf{C}} - \mathbf{C}\|_{2,\infty}$.

The second two relations follow from the Cauchy interlacing theorem below, which allows us to relate the eigenvalues of a matrix $\mathbf{M}$ and any "principal sub-matrix", which is a submatrix of $\mathbf{M}$ obtained by repeatedly removing rows and columns with the same index.

Since $\hat{\mathbf{A}} - \mathbf{A}$ and $\mathbf{A} - \hat{\mathbf{A}}$ are a principal submatrices of $\hat{\mathbf{C}} - \mathbf{C}$ and $\mathbf{C} - \hat{\mathbf{C}}$, respectively, we have $\|\hat{\mathbf{A}} - \mathbf{A}\|_2 = \max\{\lambda_{\max}(\hat{\mathbf{A}} - \mathbf{A}), \lambda_{\max}(\mathbf{A} - \hat{\mathbf{A}})\} \leq \max\{\lambda_{\max}(\hat{\mathbf{C}} - \mathbf{C}), \lambda_{\max}(\mathbf{C} - \hat{\mathbf{C}})\} = \|\hat{\mathbf{C}} - \mathbf{C}\|_2$, where the inequality follows from two applications of the interlacing theorem. Finally, since $\mathbf{A}$ is a principal submatrix of $\mathbf{C}$, we have that $\lambda_{\min}(\mathbf{A}) \geq \lambda_{\min}(\mathbf{C})$. ☐

**Theorem 3** (Cauchy Interlacing Theorem). *Let $\mathbf{M} \in \mathbb{R}^{d\times d}$ be a symmetric matrix and $\mathbf{N} \in \mathbb{R}^{r\times r}$ be a principal submatrix of $\mathbf{M}$. Then for all $j \leq r$, $\lambda_j(\mathbf{M}) \leq \lambda_j(\mathbf{N}) \leq \lambda_{d-r+j}(\mathbf{M})$, where $\lambda_j(\mathbf{M})$ denotes the $j^{\text{th}}$ smallest eigenvalue of $\mathbf{M}$. In particular, if $r = d - 1$ then*

$$\lambda_1(\mathbf{M}) \leq \lambda_1(\mathbf{N}) \leq \cdots \leq \lambda_{d-1}(\mathbf{M}) \leq \lambda_{d-1}(\mathbf{N}) \leq \lambda_d(\mathbf{M})$$

## B  Proofs for Iterative Covariance Estimation

We begin this section by formally proving that Algorithm 1 is private.

**Theorem 1.** *Algorithm 1 preserves $\left(\sum_{i=0}^d \epsilon_i\right)$-differential privacy.*

*Proof.* Let $\mathbf{X} = [\mathbf{Z}; x]$ and $\tilde{\mathbf{X}} = [\mathbf{Z}; \tilde{x}]$ be two neighboring data matrices in $\mathbb{R}^{d \times n}$ with common columns given by $\mathbf{Z} \in \mathbb{R}^{d \times (n-1)}$. For any matrix $\mathbf{A}$, let $\Lambda(\mathbf{A}) = [\lambda_1(\mathbf{A}), \dots, \lambda_d(\mathbf{A})]$ denote the vector of eigenvalues of $\mathbf{A}$. First, we argue that $\|\Lambda(\mathbf{ZZ}^\top) - \Lambda(\mathbf{XX}^\top)\|_1 \leq 1$. For any vector $v \in \mathbb{R}^d$, we have $v^\top \mathbf{XX}^\top v = v^\top \mathbf{ZZ}^\top v + v^\top x x^\top v \geq v^\top \mathbf{ZZ}^\top v$. From this, it follows that $\lambda_i(\mathbf{XX}^\top) \geq \lambda_i(\mathbf{ZZ}^\top)$ for all $i \in [d]$. Therefore, we have

$$\|\Lambda(\mathbf{XX}^\top) - \Lambda(\mathbf{ZZ}^\top)\|_1 = \mathrm{tr}(\mathbf{XX}^\top) - \mathrm{tr}(\mathbf{ZZ}^\top) = \mathrm{tr}(x x^\top) \leq 1.$$

This bound also holds for $\tilde{\mathbf{X}}\tilde{\mathbf{X}}^\top$, so $\|\Lambda(\mathbf{XX}^\top) - \Lambda(\tilde{\mathbf{X}}\tilde{\mathbf{X}}^\top)\|_1 \leq 2$. Therefore, the instance of the Laplace mechanism in step (1) preserves $\epsilon_0$-differential privacy.

Next, for any matrix $\mathbf{A}$, let $g(\mathbf{A}, \theta) = \theta^\top \mathbf{A} \theta$. For any direction $\theta \in \mathcal{S}^{d-1}$, we have

$$|g(\mathbf{XX}^\top, \theta) - g(\tilde{\mathbf{X}}\tilde{\mathbf{X}}^\top, \theta)| = |\theta^\top (x x^\top - \tilde{x}\tilde{x}^\top)\theta| \leq 2.$$

This bound also holds whenever we project the data into any subspace, so it follows that step (a) of the algorithm is using the exponential mechanism to maximize a utility function of sensitivity 2 and preserves $\epsilon_i$-differential privacy.

The final privacy guarantee follows by basic composition over the $d + 1$ queries. $\square$

We now turn to the utility guarantees of this algorithm. Starting with the proof of Lemma 5. We begin with a more precise bound on the probability that the exponential mechanism outputs a vector that captures $\tau$ less variance/energy of the covariance matrix $\mathbf{C}$ than the top eigenvector. In this result, we take advantage of the following lower bound on the $\mu_\circ$-measure of a spherical cap:

**Lemma 7** (Lemma 2.3 of Ball [1997]). *For any radius $0 \leq r \leq 2$ and any center $v \in \mathcal{S}^{d-1}$, we have*

$$\mu_\circ \left( \{ u \in \mathcal{S}^{d-1} : \|v - u\|_2 \leq r \} \right) \geq \frac{1}{2}(r/2)^{d-1}.$$

**Lemma 8.** *Let $\mathbf{X}$ be any data matrix and $\mathbf{C} = \mathbf{XX}^\top$. Let $\hat{u}$ be a sample drawn from the distribution with density proportional to $f_{\mathbf{C}}(u) = \exp(\frac{\epsilon}{4} u^\top \mathbf{C} u)$ on $\mathcal{S}^{d-1}$. For any $\tau > 0$, we have*

$$\Pr(\hat{u}^\top \mathbf{C} \hat{u} \leq \lambda_1(\mathbf{C}) - \tau) \leq 2 \exp\left(-\frac{\tau\epsilon}{8}\right) \left(\frac{8\lambda_1(\mathbf{C})}{\tau}\right)^{d-1}.$$

*Proof.* Define $G_\tau = \{ u \in \mathcal{S}^{d-1} : u^\top \mathbf{C} u \geq \lambda_1(\mathbf{C}) - \tau \}$ to be the set of directions with at most $\tau$ suboptimality. Applying the utility guarantee of Lemma 4, together with the fact that the sensitivity of $g(\mathbf{X}, u) = u^\top \mathbf{XX}^\top u$ is 2 (see the proof of Theorem 1), we have that $\Pr(\hat{u} \notin G_\tau) \leq \exp(\frac{\tau\epsilon}{8})/\mu_\circ(G_{\tau/2})$. Our main arguments focus on lower bounding $\mu_\circ(G_{\tau/2})$.

Our strategy is to pick any direction $u^* \in \arg\max_{u \in \mathcal{S}^{d-1}} u^\top \mathbf{C} u$ and argue that all directions in a spherical cap of $\ell_2$-radius $r = \frac{\tau}{4\lambda_1(\mathbf{C})}$ around $u^*$ have utility at least $\lambda_1(\mathbf{C}) - \tau/2$. It follows that $\mu_\circ(G_{\tau/2})$ is at least the $\mu_\circ$-measure of a spherical cap of $\ell_2$ radius $r = \frac{\tau}{4\lambda_1(\mathbf{C})}$, which by Lemma 7 is at least $\frac{1}{2}(\frac{\tau}{8\lambda_1(\mathbf{C})})^{d-1}$. Therefore, we have $\Pr(\hat{u} \notin G_\tau) \leq 2\exp(-\frac{\tau\epsilon}{8})(\frac{8\lambda_1(\mathbf{C})}{\tau})^{d-1}$, as required. It remains to prove the following claim:

**Claim.** *Let $u^* \in \arg\max_{u \in \mathcal{S}^{d-1}} u^\top \mathbf{C} u$. Then for every $u \in \mathcal{S}^{d-1}$ with $\|u - u^*\|_2 \leq \frac{\tau}{4\lambda_1(\mathbf{C})}$, we have $u^\top \mathbf{C} u \geq \lambda_1(\mathbf{C}) - \tau/2$.*

*Proof.* Let $r = \|u^* - u\|_2$ be the distance between $u^*$ and $u$. We want to show that if $r \leq \frac{\tau}{4\lambda_1(\mathbf{C})}$ then $u^\top \mathbf{C} u \geq \lambda_1(\mathbf{C}) - \tau/2$. Let $\|\cdot\|_{\mathbf{C}}$ be the induced seminorm (i.e., $\|v\|_{\mathbf{C}} = \sqrt{v^\top \mathbf{C} v}$). Then we have that $u^\top \mathbf{C} u = \|u\|_{\mathbf{C}}^2$ and

$$\sqrt{\lambda_1(\mathbf{C})} = \|u^*\|_{\mathbf{C}} \leq \|u^* - u\|_{\mathbf{C}} + \|u\|_{\mathbf{C}} = \|u^* - u\|_{\mathbf{C}} + \sqrt{u^\top \mathbf{C} u},$$

which implies $\sqrt{u^\top \mathbf{C} u} \geq \sqrt{\lambda_1(\mathbf{C})} - \|u^* - u\|_{\mathbf{C}}$. Next, since $\|u^* - u\|_{\mathbf{C}} = \sqrt{(u^* - u)^\top \mathbf{C}(u^* - u)} \leq r\sqrt{\lambda_1(\mathbf{C})}$, we have $u^\top \mathbf{C} u \geq (1 - r)^2 \lambda_1(\mathbf{C}) = \lambda_1(\mathbf{C}) + r^2 \lambda_1(\mathbf{C}) - 2r\lambda_1(\mathbf{C}) \geq \lambda_1(\mathbf{C}) - 2r\lambda_1(C)$. Therefore, if $r \leq \frac{\tau}{4\lambda_1(\mathbf{C})}$, we have $2r\lambda_1(\mathbf{C}) \leq \tau/2$ and thus $u^\top \mathbf{C} u \geq \lambda_1(\mathbf{C}) - \tau/2$. $\square$

$\square$

All that remains to prove Lemma 5 is to choose an appropriate value of $\tau$.

**Lemma 5.** *Let $\mathbf{X} \in \mathbb{R}^{d \times n}$ be a data matrix and $\mathbf{C} = \mathbf{X}\mathbf{X}^\top$. For any $\beta > 0$, with probability at least $1 - \beta$ over $\hat{u}$ sampled from the density proportional to $f_{\mathbf{C}}(u) = \exp(\frac{\epsilon}{4} u^\top \mathbf{C} u)$ on $\mathcal{S}^{d-1}$, we have*

$$\hat{u}^\top \mathbf{C} \hat{u} \geq \lambda_1(\mathbf{C}) - O\left(\frac{1}{\epsilon}\left(d \log \lambda_1(\mathbf{C}) + \log \frac{1}{\beta}\right)\right)$$

*Proof.* The proof follows by setting $\tau = \frac{8}{\epsilon} \log\left(e + \frac{2(8\lambda_1(\mathbf{C}))^{d-1}}{\beta}\right)$ in Lemma 8. In particular, this guarantees that with probability at least $1 - \beta$, we have

$$\hat{u}^\top \mathbf{C} \hat{u} \geq \lambda_1(\mathbf{C}) - \frac{8}{\epsilon} \log\left(e + \frac{2(8\lambda_1(\mathbf{C}))^{d-1}}{\beta}\right).$$

$\square$

We can now prove the main utility guarantee of our algorithm.

**Theorem 2.** *Let $\hat{\mathbf{C}}$ be the output of Algorithm 1 run with inputs $\mathbf{C}$ and privacy parameters $\epsilon_0, \ldots, \epsilon_d$. For any $\beta > 0$, with probability at least $1 - \beta$ we have*

$$\|\mathbf{C} - \hat{\mathbf{C}}\|_F \leq \tilde{O}\left(\sqrt{\sum_{i=1}^d \frac{d\lambda_i(\mathbf{C})}{\epsilon_i}} + \frac{\sqrt{d}}{\epsilon_0}\right),$$

*where the $\tilde{O}$ notation suppresses logarithmic terms in $d$, $\lambda_1(\mathbf{C})$, and $\beta$.*

*Proof.* First we bound the error in each of the differentially private estimations made by Algorithm 1. Let $\beta' = \beta/(2d)$. For each eigenvalue estimate, Lemma 3 guarantees that with probability at least $1 - \beta/2$, we have $|\hat{\lambda}_i - \lambda_i(\mathbf{C})| \leq \frac{2}{\epsilon_0} \log \frac{1}{\beta'}$. For each eigenvector estimate, Lemma 5 guarantees that with probability at least $1 - \beta'$ we have

$$\lambda_1(\mathbf{C}_i) - \hat{u}_i^\top \mathbf{C}_i \hat{u}_i \leq O\left(\frac{1}{\epsilon_i}((d - i + 1)\log \lambda_1(\mathbf{C}_i) + \log \frac{1}{\beta'})\right). \tag{2}$$

By the union bound, all events hold simultaneously with probability at least $1 - \beta$. Assume this high probability event holds for the remainder of the proof.

The key remaining step is to argue that the guarantee from (2) implies that $\lambda_i(\mathbf{C}) - \hat{\theta}_i^\top \mathbf{C} \hat{\theta}_i$ is also small. First, let $u \in \mathbb{R}^{d-i+1}$ be any unit vector and let $\theta = \mathbf{P}_i^\top u$. Then we have $\|\theta\|_2 = 1$ (since $\mathbf{P}_i$ has orthonormal rows) and $\theta^\top \mathbf{C}\theta = u^\top \mathbf{P}_i \mathbf{C} \mathbf{P}_i^\top u = u^\top \mathbf{C}_i u$. Additionally, if a unit vector $\theta \in \mathbb{R}^d$ can be expressed as $\theta = \mathbf{P}_i^\top u$, then $\|u\|_2 = 1$, and $\theta$ belongs to the $(d-i+1)$-dimensional subspace spanned by the rows of $\mathbf{P}_i$. Thus, using the min-max characterization of $\lambda_i(\mathbf{C})$ we have

$$\lambda_i(\mathbf{C}) = \min_{\substack{V \subset \mathbb{R}^d \\ \dim(V) = d-i+1}} \max_{\substack{\theta \in V \\ \|\theta\|_2 = 1}} \theta^\top \mathbf{C}\theta \leq \max_{\substack{\theta \in \mathrm{rowspan}(\mathbf{P}_i) \\ \|\theta\|_2 = 1}} \theta^\top \mathbf{C}\theta = \max_{\substack{u \in \mathbb{R}^{d-i+1} \\ \|u\|_2 = 1}} u^\top \mathbf{C}_i u = \lambda_1(\mathbf{C}_i).$$

This implies that

$$\lambda_i(\mathbf{C}) - \hat{\theta}_i^\top \mathbf{C} \hat{\theta}_i \leq \lambda_1(\mathbf{C}_i) - \hat{u}_i^\top \mathbf{C}_i \hat{u}_i \leq O\left(\frac{1}{\epsilon_i}(d \log \lambda_1(\mathbf{C}) + \log \frac{1}{\beta'})\right),$$

where the final inequality uses (2) together with the fact that $\lambda_1(\mathbf{C}_i) \leq \lambda_1(\mathbf{C})$. Combining with Lemma 6, we have

$$\|\mathbf{C} - \hat{\mathbf{C}}\|_F \leq O\left(\sqrt{\sum_{i=1}^d \lambda_i(\mathbf{C})(\frac{1}{\epsilon_i}(d \log \lambda_1(\mathbf{C}) + \log \frac{1}{\beta'})} + \frac{\sqrt{d}\log \frac{1}{\beta'}}{\epsilon_0}\right)$$

$$= \tilde{O}\left(\sqrt{\sum_{i=1}^d \frac{d\lambda_i(\mathbf{C})}{\epsilon_i}} + \frac{\sqrt{d}}{\epsilon_0}\right),$$

as required. $\square$

Next we prove the error bound for Algorithm 1 using the adaptive privacy budgeting.

**Corollary 1.** *Fix any privacy parameter $\epsilon$ and any failure probability $\beta > 0$, let $\epsilon_0 = \epsilon/2$, and let $\epsilon_i = \frac{\frac{\epsilon}{2}\sqrt{\hat{\lambda}_i + \tau}}{\sum_j \sqrt{\hat{\lambda}_j + \tau}}$ where $\tau = \frac{2}{\epsilon_0}\log(2d/\beta)$. Then Algorithm 1 run with $\epsilon_0, \ldots, \epsilon_d$ preserves $\epsilon$-differential privacy and, with probability at least $1 - \beta$, the output $\hat{\mathbf{C}}$ satisfies*

$$\|\mathbf{C} - \hat{\mathbf{C}}\|_F \leq \tilde{O}\left(\sqrt{\frac{d}{\epsilon}}\sum_{i=1}^{d}\sqrt{\hat{\lambda}_i + \frac{1}{\epsilon}} + \frac{\sqrt{d}}{\epsilon}\right).$$

*Proof.* With probability at least $1 - \beta$ we are guaranteed that the high probability event from Theorem 2 occurs. In particular, we have that $|\hat{\lambda}_i - \lambda_i(C)| \leq \tau$ for all $i$ and that

$$\|\mathbf{C} - \hat{\mathbf{C}}\|_F \leq \tilde{O}\left(\sqrt{\sum_{i=1}^{d}\frac{d\lambda_i(\mathbf{C})}{\epsilon_i}} + \frac{\sqrt{d}}{\epsilon_0}\right).$$

Assume this high probability event holds for the remainder of the proof.

Given that $\lambda_i(\mathbf{C}) \leq \hat{\lambda}_i + \tau$ and the above bound is increasing in $\lambda_i(\mathbf{C})$, we can upper bound the loss in terms of the estimated eigenvalues as well:

$$\|\mathbf{C} - \hat{\mathbf{C}}\|_F \leq \tilde{O}\left(\sqrt{\sum_{i=1}^{d}\frac{d(\hat{\lambda}_i + \tau)}{\epsilon_i}} + \frac{\sqrt{d}}{\epsilon_0}\right).$$

Substituing the given values for $\epsilon_0, \ldots, \epsilon_d$, this becomes

$$\|\mathbf{C} - \hat{\mathbf{C}}\|_F \leq \tilde{O}\left(\sqrt{\sum_{i=1}^{d}d(\hat{\lambda}_i + \tau) \cdot \frac{2\sum_j \sqrt{\hat{\lambda}_j + \tau}}{\epsilon\sqrt{\hat{\lambda}_i + \tau}}} + \frac{\sqrt{d}}{\epsilon}\right)$$

$$= \tilde{O}\left(\sqrt{\frac{d}{\epsilon}\sum_{i=1}^{d}\sqrt{\hat{\lambda}_i + \tau}\sum_{j=1}^{d}\sqrt{\hat{\lambda}_j + \tau}} + \frac{\sqrt{d}}{\epsilon}\right)$$

$$= \tilde{O}\left(\sqrt{\frac{d}{\epsilon}\left(\sum_{i=1}^{d}\sqrt{\hat{\lambda}_i + \tau}\right)^2} + \frac{\sqrt{d}}{\epsilon}\right)$$

$$= \tilde{O}\left(\sqrt{\frac{d}{\epsilon}}\sum_{i=1}^{d}\sqrt{\hat{\lambda}_i + \tau} + \frac{\sqrt{d}}{\epsilon}\right),$$

as required. $\qquad\square$

**Corollary 2.** *For any $\zeta > 0$ and any positive semidefinite matrix $\hat{\mathbf{C}}$, with probability at least $0.99$ (or any fixed success probability), running Algorithm 1 with $\epsilon_0 = \epsilon/2$ and $\epsilon_i = \epsilon/(2d)$ for $i = 1, \ldots, d$ preserves $\epsilon$-differential privacy and outputs $\hat{\mathbf{C}}$ such that $\|\mathbf{C} - \hat{\mathbf{C}}\|_F \leq O(\zeta\lambda_1(\mathbf{C}))$ if $\lambda_1(\mathbf{C}) \geq \frac{2d^3}{\epsilon\zeta^2}\log(\frac{d}{\epsilon\zeta})$.*

*Proof.* From Theorem 2, we know that with probability at least $0.99$ the output $\hat{\mathbf{C}}$ of Algorithm 1 satisfies

$$\|\mathbf{C} - \hat{\mathbf{C}}\|_F \leq O\left(\sqrt{\sum_{i=1}^{d}\frac{d\lambda_i(\mathbf{C})\log(\lambda_1(\mathbf{C}))}{\epsilon_i}} + \frac{\sqrt{d}}{\epsilon_0}\right).$$

Upper bounding $\lambda_i(\mathbf{C})$ by $\lambda_1(\mathbf{C})$ and substituting the given privacy parameters $\epsilon_0, \ldots, \epsilon_d$, we have

$$\|\mathbf{C} - \hat{\mathbf{C}}\|_F \leq O\left(d^{3/2}\sqrt{\lambda_1(\mathbf{C})\log(\lambda_1(\mathbf{C}))/\epsilon} + \sqrt{d}/\epsilon\right)$$

$$\leq O\left(\max\{d^{3/2}\sqrt{\lambda_1(\mathbf{C})\log(\lambda_1(\mathbf{C}))/\epsilon}, \sqrt{d}/\epsilon\}\right)$$

---
**Algorithm 2** Top Eigenvector Sampler
---
**Input:** Covariance matrix $\mathbf{C} \in \mathbb{R}^{d \times d}$, privacy parameter $\epsilon > 0$.
1. Let $\mathbf{A} = -\frac{\epsilon}{4}\mathbf{C} + \frac{\epsilon}{4}\lambda_d(\mathbf{C}) \cdot I$.

2. Let $\mathbf{\Omega} = I + 2\mathbf{A}/b$ where $b$ satisfies $\sum_{i=1}^{d} 1/(b + 2\lambda_i(\mathbf{A})) = 0$.

3. Let $M = \exp(-(d-b)/2) \cdot (d/b)^{d/2}$.

4. Repeat forever:
   (a) Sample $z \sim \mathcal{N}(0, \Omega^{-1})$ and set $u = z/\|z\|_2$.
   (b) With probability $\frac{\exp(-u^\top \mathbf{A} u)}{M \cdot (u^\top \mathbf{\Omega} u)^{d/2}}$ 'accept' and return $u$.
---

---
**Algorithm 3** Data generating algorithm
---
**Input:** Dimension $d$, number of points $n$.
1. Sample $\mathbf{X} \in \mathbb{R}^{d \times n}$ from normal distribution $N(0,1)$. Sample $\mathbf{U} \in \mathbb{R}^{d \times d}$ from uniform distribution $U(0,1)$.

2. Return Normalize($\mathbf{UX}$)
---

It follows that if $\zeta\lambda_1(\mathbf{C}) \geq \max\{d^{3/2}\sqrt{\lambda_1(\mathbf{C})\log(\lambda_1(\mathbf{C}))/\epsilon}, \sqrt{d}/\epsilon\}$, then $\|\mathbf{C} - \hat{\mathbf{C}}\|_F \leq O(\zeta\lambda_1(\mathbf{C}))$. Whenever $\lambda_1(\mathbf{C}) \geq \sqrt{d}/(\epsilon\zeta)$ we have $\zeta\lambda_1(\mathbf{C}) \geq \sqrt{d}/\epsilon$. On the other hand, if $\lambda_1(\mathbf{C}) \geq \frac{d^3}{\epsilon\zeta^2}\log(\lambda_1(\mathbf{C}))$, we have that $\zeta\lambda_1(\mathbf{C}) \geq d^{3/2}\sqrt{\lambda_1(\mathbf{C})\log(\lambda_1(\mathbf{C}))/\epsilon}$. Using the fact that for any $a > 0$, the inequality $x \geq 2a\log(a)$ implies that $x \geq a\log(x)$, it follows that when $\lambda_1(\mathbf{C}) \geq \frac{2d^3}{\epsilon\zeta^2}\log(\frac{d}{\epsilon\zeta})$ we have that $\zeta\lambda_1(\mathbf{C}) \geq d^{3/2}\sqrt{\lambda_1(\mathbf{C})\log(\lambda_1(\mathbf{C}))/\epsilon}$. Since the second requirement on $\lambda_1(\mathbf{C})$ is stronger, we are guaranteed that when $\lambda_1(\mathbf{C}) \geq \frac{2d^3}{\epsilon\zeta^2}\log(\frac{d}{\epsilon\zeta})$, with probability at least 0.99 we have $\|\mathbf{C} - \hat{\mathbf{C}}\|_F \leq O(\zeta\lambda_1(\mathbf{C}))$. $\qquad\square$

## C  Experiments

In this section we provide a more comprehensive empirical evaluation of our algorithm.

### C.1  Synthetic datasets

To properly measure the effects of dimension and data set size we evaluate the performance of all algorithms on a synthetic data set. The data is generated according to Algorithm 3, where Normalize() is a subroutine that ensures every column of $\mathbf{X}$ has mean 0, variance 1, and $L_2$-norm 1. Intuitively, the uniform matrix $\mathbf{U}$ described in the algorithm is used to introduce correlations between between features.

We consider different data regimes by varying $d \in \{10, 100\}$ and $n \in \{1000, 10000, 50000\}$. Similar to Section 4 we measure the error of the algorithms by the normalized Frobenius distance between the estimated and true covariance matrices. We ran each experiment 100 times and the average error is presented in Figure 2.

The results of these experiments are fairly similar to the ones on the real data sets. Our algorihtm seems to consistently outperform all others. What is truly interesting however is to observe how the performance gap between **AD** and **KT** changes as a function of $d$ and $n$. As the dimension increases the gap seems to increase and the opposite occurs as $n$ increases. This was already foretold in Section 3.1. However, by observing this gap empirically we show that the theoretical improvement is not an artifact of the analysis but a an inherent quality of the algorithm. When comparing against the Gaussian mechanism we see that the relative performance of our algorithm degrades as $n$ increases. This was also established theoretically in Section 3.1. On the other hand, it is interesting to observe that for high privacy regimes, the relative performance of our algorithm is better as the dimension increases. This phenomenon can be explained by the fact that the utility guarantees of the Gaussian mechanism are in $O(\frac{d^{3/2}}{\epsilon})$ whereas those of our algorithm are in $O(\frac{d^{3/2}}{\sqrt{\epsilon}} + \frac{d}{\epsilon})$. Thus our performance will be better for small values of $\epsilon$ and large values of $d$.

Figure 2: (a) Comparison of our algorithm to other $\epsilon$-differentially private algorithms on artificial data. Rows in the grid correspond to a fixed value of $d$, while columns correspond to a fixed value of $n$. (b) Comparison of our algorithm to the Gaussian mechanism. G-$x$ corresponds to the Gaussian mechanism with parameter $\delta = 10^{-x}$

## C.2 Regression

We now turn to the problem of learning a regressor in a private manner. More precisely, we will attempt to predict the first coordinate of the data matrix $\mathbf{X}$ based on all other coordinates. Let $\mathbf{X}_{s:t}$ denote a matrix consisting of columns of matrix $\mathbf{X}$ in the range $[s, t]$. Let $\mathbf{y} = \mathbf{X}_{1:1}$ and $\mathbf{X}' = \mathbf{X}_{2:d}$. We are interested in solving the problem

$$\min_{\mathbf{w} \in \mathbb{R}^d} \|\mathbf{X}'\mathbf{w} - \mathbf{y}\|^2 + \alpha \|\mathbf{w}\|^2. \tag{3}$$

while ensuring that the output is private.

One way to solve this problem is to use the private covariance matrix to find the optimal vector $\mathbf{w}$, as shown in Section 1.1. A different approach is to solve (3) on the true data and then add noise to the

Figure 3: Results comparing our algorithm to the perturbation algorithm of Chaudhuri et al. [2011] on the regression task. The value of the regularization parameter is appended to the algorithm name in the plot legend. The error metric is the average squared loss when predicting the first coordinate of the data.

output, as described in the output perturbation algorithm of Chaudhuri et al. [2011]. We refer to this perturbation algorithm as **P**. We refrain from comparing to objective perturbation methods such as those of Chaudhuri et al. [2011] and Kifer et al. [2012] since their performance is somewhat similar to that of output perturbation. Moreover, our goal is not to provide a state of the art algorithm for regression. Instead, we want to show that the utility of our algorithm does not degrade too much compared to a specialized algorithm, even when the privacy task of releasing a full covariance matrix is harder than that of releasing a regression vector.

To assess the quality of the solution $\mathbf{w}$ we measure the squared loss of the prediction on a test data set generated from the same distribution as $\mathbf{X}$. Since the baseline method of Chaudhuri et al. [2011] depends on the regularization parameter $\alpha$, we vary this parameter over the set $\{0.1, 1, 10, 100, 1000\}$. The results of this comparison on synthetic data are presented in Figure 3.

We begin by noting that, for a fixed $\epsilon$, the best performing version of **P** performs similarly to the best performing version of **AD** (although there are cases where each algorithm performs slightly better). Nonetheless, our algorithm has the practical advantage that its performance is much more stable across different choices of $\alpha$.

The instability of **P** can be understood from the fact that $\alpha$ plays two roles in the algorithm of Chaudhuri et al. [2011]. First, it serves as a regularization parameter that helps avoid overfitting. Second, it works as a privacy parameter: their approach adds less noise for larger $\alpha$. While this may be a desirable property, it introduces a complication, since $\alpha$ must be tuned privately for optimal performance. As an example, consider the case when $d = 10$ and $n = 10000$. For our algorithm **AD**, the optimal regularization parameter is fixed at $\alpha = 100$ for all $\epsilon$. For the **P** baseline, the performance of a fixed $\alpha$ degrades rapidly as $\epsilon$ decreases, and different values of $\alpha$ emerge as optimal. Since performing this tuning requires adding a different amount of noise every time, each tuning iteration uses additional privacy budget. In contrast, since we release a single covariance matrix, we can test an unlimited number of values for $\alpha$ and remain differentially private without any additional privacy loss.

Another advantage of privately estimating the full covariance matrix is that it allows us to regress any coordinate against any of the others. This is not true of the perturbation algorithm, which operates one instance at a time. In their setting, if we wanted to choose different regression targets we would have to split the privacy budget $\epsilon$ among all of the runs. To show how privacy splitting affects the

Figure 4: Results comparing our algorithm against the perturbation algorithm of Chaudhuri et al. [2011] on the regression task. The error metric is the average squared loss across all possible targets in the data matrix.

Figure 5: Comparison to the perturbation algorithm of Chaudhuri et al. [2011]. Error is the squared loss on the test data.

perturbation algorithm, we conducted the same regression task $d$ times to predict each column of the data matrix $\mathbf{X}$ from the others. In every run we set the privacy parameter in the perturbation algorithm to $\frac{\epsilon}{d}$. We show the average squared loss (over all predicted columns) in Figure 4. Clearly, if there is a need to perform multiple regressions on the same dataset, our approach offers significant advantages.

We conclude this section by solving the same regression task on the real-world datasets used in Section 4. We splitted the data set and used 80% for training and 20% for testing. The results shown in Figure 5 report the test error.