[Reviews · NeurIPS 2019]

Reviewer 1



Content: In this work, they design an efficient pure differentially private algorithm for covariance matrix estimation. Their empirical results show that their algorithm outperforms several other algorithms in the literature in practice. Their theoretical results imply that their algorithm outperforms simple noise addition in the high privacy/small dataset setting. They explicitly consider the sampling procedure for implementing the exponential mechanism in their setting. I appreciated this part of the paper since this is often swept under the rug. The highlight the problems associated to relying on Gibbs sampling and instead use a rejection sampling scheme proposed by Kent et al. Originality: The authors’ highlight several ways that their work differs from previous work. Previous work seems to focus on either approximate DP or sparse matrices. In contrast, this paper focuses on pure DP and attempts to estimate the entire covariance matrix with no sparsity assumption. The closest work seems to be Kapralov and Talwar, who (as far as I understand) also use an exponential mechanism and peeling technique. This work uses a different peeling technique to that work, when a new eigenvector is chosen, the authors’ project onto the orthogonal space rather than subtracting the new component of the eigendecomposition as in KT. The authors’ claim that this change allows them to implement a more efficient sampling mechanism. They also show in Corollary 2 that it can result in a more accurate estimate. I don’t know enough about the DP covariance estimation literature to say if the algorithms they tested against represent a full spectrum of DP covariance estimations algorithms. Their algorithm seems to The claim that this is the “first comparative empirical evaluation of different private regression methods” seems incorrect. Their are several others including, but not limited to, “Revisiting differentially private linear regression: optimal and adaptive prediction & estimation in unbounded domain” by Yu-Xiang Wang which also compares several linear regression methods on datasets from the UCI repository. The authors’ are not claiming that their algorithm should outperform all other algorithms in the linear regression setting since it is not tailored to this purpose (it seems inevitable that releasing a general purpose statistic will come at some cost), although it would be interesting to see how it compares to the more promising algorithms presented by Wang. Quality: The proofs seem complete (although I didn’t read the appendix in detail). I appreciate that they benchmarked against other algorithms on three different datasets. It would be nice to have some description of the properties of these datasets and why they were chosen. Are they datasets that differ significantly enough to capture a wide range of behaviors of the algorithm? Clarity: The paper is well-written (bar a few typos) and easy to follow. The main algorithm is clearly presented. The experiments are clearly described. A minor comment since it is in the appendix, but I would like to see them compare the linear regression algorithm to a few more benchmark algorithms. The algorithms they have don’t seem to capture the full range of DP linear regression algorithms that currently appear in the literature. Significance: Given the popularity of linear regression and PCA, better algorithms for performing these tasks are an important research direction. Many of the DP algorithms for these simple data analysis tasks have focuses on the big data/ low privacy regimes. However, since there is a lot of research in the social science world that works with small datasets, it is important to develop careful algorithms that perform well for these datasets. The authors’ make good progress in this direction. In this vein, as more practitioners implement DP mechanisms, it is important to have efficient implementations of DP algorithms and explore the performance of these algorithms on real datasets. The different mechanisms perform quite differently on the three different datasets. The AD algorithms is the best almost across the board but the shape of the performance curves varies significantly. This phenomenon, that the data structure plays a significant role in the comparison between algorithms, also occurs in DP linear regression. I’d be interested to see further work attempting to clarify the conditions of the data which are necessary for AD to be the optimal algorithm. For example, what do the authors’ conjecture is happening in the wine data set with eps=0.01? Minor comments: - Typo: introduction, eps, delta-> (eps, delta) - In line 14, “attack vectors” is an uncommon term - The quote “These algorithms typically train directly on the raw data… standard off-the-shelf algorithms for learning” in lines 20-23 doesn’t feel strictly true. I think the author is trying to make the point that sometimes it is useful to release summary statistics (I.e. synthetic data) that can be used for a wide variety of data analysis purposes. This is indeed a valuable approach, however it also has pitfalls which make me wary of calling it a “more general alternative approach”. - I’m unsure what the authors’ mean by optimal in line 58. They derive a better than trivial way to allocate the privacy budget to each of the eigenvectors, although it’s not clear to me that this is optimal. - The notation x(-t) is confusing since it is of a different type to x(t). In general, the presentation of the notation for linear regression seems more confusing than it needs to be. - If the data truly is sparse, do the authors’ conjecture that one should use a technique like Chaudhuri et al. or Wang and Xu, that is tailored to sparse data? - Overloaded notation: \tilde{X} is used in both line 83 and line 134 for different purposes. - Typo? In Algorithm 1, should it be \hat{theta_i) = P_i^T\hat{u_i}? - Typo: Algorithm 1: ortohonormal-> orthonormal - Overloaded notation: delta is used as a privacy parameter and also in Lemma 5 as a failure probability. Perhaps use beta instead for the failure probability. - I’m a little confused by the comments in lines 279-281 regarding the expected number of rejections. Why is it the case that the observed number of samples is much greater than the bound given in Kent et al., do the conditions not hold? - The graphs are quite small. This is understandable in the main paper since space is limited but it would be nice to have them bigger in the appendix.

Reviewer 2



Pros: * The new algorithm scales better with the dimension and eps: their dependence is d^{3/2} / sqrt(eps) vs d^{3/2} / eps for instance for Analyze Gauss. I think this scaling is very interesting, as it shows that when the dimensionality of the data is high (and for reasonably small eps), this method is to be preferred. The authors do not emphasize this very much for some reason, but I think it is a nice property of their algorithm. * The algorithm, while only subtly different than the on in Kapralov-Talwar, seems substantially more natural: projection does feel to be the more natural operation than subtracting out the eigenvector, for exactly the reasons the authors mention. This is also similar to the intuition behind the private recursive preconditioning in [1]. It would be interesting to see if there is any technical connection between these two methods, as they seem quite similar. Cons: * The new algorithm improves upon Analyze Gauss in only a relatively small set of parameter regimes. When n is somewhat large, in particular, the new guarantees are worse than that of Analyze Gauss. * The assumption that the columns have bounded l2 norm is quite restrictive and somewhat unrealistic. For instance, most natural generative models have columns that would be scaling with the dimension, at least. It would interesting if these methods could also depend more strongly on the geometry of the data points, for instance, if the data is ill-conditioned. * When the data matrix is low rank (or approximately low rank), it appears that this algorithm gives weak guarantees. In particular, the very first step of estimating the eigenvalues to additive O(1/eps_0) could already plausibly destroy any meaningful low-rank structure. As a result, they can only achieve Frobenius-style approximation guarantees, which don't say much when the matrix is low rank. Despite these flaws, I still think the paper is interesting enough to merit acceptance. [1] G. Kamath, J. Li, V. Singhal, J. Ullman. Privately learning high dimensional distributions. COLT 2019.

Reviewer 3



Originality: The proposed algorithm and analysis are very similar to [Kapralov and Talwar 2013]. The efficient implementation of sampling a vector on unit sphere utilizes a more recent work of [Kent et al 2018]. The main change is algorithmic: instead of subtracting rank-1 matrix from exponential mechanism each step, the new algorithm projects the data so that it is orthogonal to this rank-1 subspace. Though the algorithm and analysis are similar to [Kapralov and Talwar 2013], it requires non-trivial amount of work to show utility analysis. The outline of proof is similar - proving utility of eigenvectors (seeing the difference of lambda_i(C) and theta^hat C theta^hat) and eigenvalues (bounding tau in Lemma 6). The paper also shows the first empirical comparison of known DP covariance estimation algorithms. Related work and preliminary section cover main previous work and yet are not overly verbose. Quality: The paper is well written. The proofs and details are complete. The experiments are extensive and their results are adequately discussed. Small comments: - Lemma 1 has a utility bound with RHS depending on w^hat_alpha. I would think that RHS should depends only on C,C^hat (which is why you want to estimate covariance). It is a little disappointing - or is there an explanation why depending on regression coefficient is needed? - line 95 formatting typo: the second C^hat is not bold --> should be bold - line 1 of Alg 1: "C_1 = C" is repeated twice - I understand that the main point of theoretical contribution is to have pure-DP (Theorem 1), but wouldn't it be natural to also do advanced composition or others to get (eps,delta)-DP and check if it improves the privacy analysis, compared to other (eps,delta)-DP? At least can include this in Appendix if it does not help. - line 230, forget a space after the full stop. Clarity: The paper is clear and concise - easy to read. The introduction is concise with good overview of subject, motivation, their algorithm's quick explanation, and contributions. Clear comparison (e.g. Section 3.2) make their result easier to understand in the context. Small comments: - it may be beneficial to readers why/if tuning eps_i helps the algorithm or not after Corollary 1. It is mentioned as one line on empirical result section (line 317) that it does not change the performance significantly, but is there any intuition why/when/what type of data this may help? - line 300-301: what about eps_0 that is used to estimate the eigenvalues? Same as those eps_i for eigenvectors estimation? Significance: Though similar to previous algorithm of [Kapralov and Talwar 2013], their new algorithm is significantly different enough with different proven privacy and utility bounds. The empirical results is promising. The only few details missing are running time (how it scale to real datasets and compared to other algorithms), especially to convince readers that Algorithm 2 (only non trivial part of their algorithm, in my opinion) is not hard to implement and is efficient.

[Author Response · NeurIPS 2019]

Thank you to all the reviewers for their detailed comments. We will make sure to address all minor feedback in the final version of the paper.

**R1**

- *What do the authors conjecture is happening in the wine data set with $\epsilon = 0.01$?* For small datasets like wine, when using small values of epsilon the output contains almost no information about the input. Instead, utility is determined mostly by coincidental alignment between the mechanisms noise distribution and the specific dataset at hand.

- *If the data truly is sparse, do the authors conjecture that one should use a technique like Chaudhuri et al. or Wang and Xu, that is tailored to sparse data?* Yes, if there is public knowledge that the matrix will be sparse, one should use the techniques of Wang and Xu. However, it is worth mentioning that in practice covariance matrices are generally not sparse (although they may be low rank).

- *Why is it the case that the observed number of samples is much greater than the bound given in Kent et al.?* This is probably explained by the fact that the scale of $d$ is not large enough. Kent et al. provide only an asymptotic bound on the number of samples.

- *Im unsure what the authors mean by optimal in line 58.* We meant that the parameters optimize our error bound, but youre right that this is not optimal in general, and the wording is misleading. We will make this clearer in the final version of the paper.

**R2**

- *When n is somewhat large, in particular, the new guarantees are worse than that of Analyze Gauss.* As touched upon in the introduction, we view large n as coinciding with low sensitivity and therefore easily attainable privacy. Indeed, for a fixed epsilon as n becomes large all reasonable mechanisms begin to perform well, and the theoretical distinctions vanish in practice. In contrast, practitioners can face difficulties when attempting to deploy mechanisms on hundreds or thousands of examples, for instance in the social sciences or medicine. We will make this fact more clear in the paper.

- *The assumption that the columns have bounded $L_2$ norm is quite restrictive and somewhat unrealistic.* We emphasize that a bound $B$ on the column norm can easily be surfaced to the theorems (rather than assumed equal to one), and will do so in the final version of the paper. Some kind of scale dependence is, of course, fundamental. We agree that an analysis incorporating the conditioning of the matrix could make for interesting future work.

- *When the data matrix is low rank (or approximately low rank), it appears that this algorithm gives weak guarantees.* If the rank $k$ is known beforehand then our algorithm is easily adapted by terminating after $k$ (rather than $d$) iterations. Furthermore, rank is 1-sensitive and therefore can be easily estimated in a differentially private manner by adding Laplace noise to the rank of the true matrix.

**R3**

- *Lemma 1 has a utility bound with RHS depending on $\widehat{w}_\alpha$* A dependence on either $\|w_\alpha\|$ or $\widehat{w}_\alpha\|$ is necessary since, intuitively, the scale of $w_\alpha$ will affect the scale of the error. The easiest way to see this formally is to inspect the equations after line 399 in the Appendix. The bound is in fact symmetric, and can be written in terms of $\|w_\alpha\|$; however, we chose the stated bound because $\|\widehat{w}_\alpha\|$ can be computed from the private matrix and therefore is known to the practitioner.

- *Publicly available code may help the significance of this paper.* We will be releasing an open source version of our code soon.

[Meta-Review · NeurIPS 2019]

After a discussion considering all the reviews and author feedback, all reviewers recommend acceptance of the paper. The reviewers note a number of minor weaknesses, but despite those the paper appears to make a sufficiently interesting and useful contribution.